# STYLE2SHAPE: IMAGE STYLE GUIDED 3D SHAPE MATERIAL GENERATION

## ABSTRACT

This paper presents Style2Shape, a novel framework for generating physically-based rendering (PBR) materials for 3D models from a single reference image. Unlike existing methods limited by the diversity of procedural material libraries or producing non-editable representations, our approach combines procedural materials with generated textures via differentiable rendering. Our key insight is that procedural parameters ensure reflectance correctness while generated textures capture arbitrary appearances-their learnable combination achieves both physical plausibility and visual fidelity. The framework operates in three stages: (1) structure-guided appearance transfer that synthesizes geometrically-aligned supervision, (2) hybrid PBR material initialization that retrieves procedural materials based on physical properties and generates complementary textures for appearance details, and (3) physics-based optimization jointly refining all components through differentiable rendering. Extensive experiments demonstrate that our approach generates high-fidelity results, producing editable PBR materials that faithfully reproduce reference appearances while maintaining physical plausibility. The generated assets are structured to be compatible with standard 3D rendering workflows.

## 1 INTRODUCTION

Creating realistic materials for 3D objects is fundamental to computer graphics, with critical applications spanning architectural visualization, film production, and virtual reality. While recent advances in differentiable rendering and generative models have revolutionized 3D content creation, acquiring physically-based materials from limited observations remains a significant challenge.

Recovering physically-based rendering (PBR) materials from a single reference image is inherently ill-posed. The complex interplay between geometry, material properties, and illumination creates fundamental ambiguities—a bright surface could result from either high albedo or strong lighting. Moreover, the domain gap between arbitrary reference images and target 3D geometries prevents direct material transfer.

Current approaches exhibit distinct limitations: text-driven methods (Chen et al., 2023; Michel et al., 2022) suffer from description ambiguity, image-based methods (Liu et al., 2023; 2024) produce non-editable implicit representations, and Material Palette (Lopes et al., 2024) requires manual mapping and fails with perspective distortions. While procedural materials (Shi et al., 2020; Hu et al., 2022; Yan et al., 2023) offer physically-plausible representations, they are constrained by predefined parameter spaces and cannot capture arbitrary real-world appearances.

A more promising direction involves procedural materials (Shi et al., 2020; Hu et al., 2022), which offer physically-plausible and editable representations. PSDR-Room (Yan et al., 2023) demonstrates the power of differentiable procedural materials. However, a fundamental limitation persists: procedural materials excel at representing physically-correct reflectance properties but are inherently constrained by their predefined parameter spaces, failing to capture arbitrary real-world appearances. Conversely, generative models can synthesize novel textures but lack physical grounding.

This raises a fundamental question: Can we combine the reflectance correctness of procedural materials with the expressive power of generated textures through differentiable rendering to achieve both physically plausible and arbitrary appearance modeling?

We present a novel three-stage framework that addresses this challenge through hybrid material optimization. Our key insight is that procedural materials and generated textures are complementary: procedural parameters ensure correct reflectance properties (roughness, metallicity, etc.), while generated textures capture arbitrary visual patterns. By optimizing their learnable combination within a physics-based differentiable rendering pipeline, we achieve materials that are both physically plausible and visually faithful to reference image.

Our framework proceeds as follows:

**(1) Structure-guided Appearance Transfer:** We reformulate appearance transfer as an image-to-image translation problem, synthesizing geometrically-aligned supervision images that preserve reference appearance while matching the target model's geometry.

**(2) Hybrid PBR Material Initialization:** We retrieve procedural materials based on physical properties and generate complementary textures for visual patterns, combining them through learnable blending weights.

**(3) Physics-Based Differentiable Optimization:** We jointly optimize procedural parameters, generated textures, and their blending within a differentiable rendering framework, ensuring convergence to physically-correct and visually-accurate materials.

In summary, the main contributions of our work include:

- A unified framework for generating editable PBR materials from a single image, bridging the gap between generative models and physically-based rendering.
- A hybrid material representation that combines procedural materials for reflectance correctness with generated textures for arbitrary appearance, unified through learnable blending.
- A physics-based optimization framework that utilizes differentiable rendering to jointly optimize all material components while maintaining physical plausibility.

## RELATED WORK

### 1.1 MATERIAL GENERATION FOR 3D OBJECT

The generation of materials for 3D models has evolved significantly, moving from simple texture mapping to the synthesis of complex, physically-based rendering (PBR) materials. Early work by (Munkberg et al., 2022) pioneered a method for jointly optimizing the geometry, materials, and lighting from multi-view images, employing differentiable rendering and coordinate-based networks to represent volumetric textures for gradient-based optimization on the surface mesh.

**Text-driven Generation.** With the rise of generative models, text-driven material generation has become a prominent research area. Text2Mesh (Michel et al., 2022) established a paradigm for text-driven stylization by predicting colors and local geometric details for a 3D mesh based on a text prompt. Recent works enhance physical realism through various approaches: Fantasia3D (Chen et al., 2023) disentangles geometry from appearance, MATLABER (Xu et al., 2023) uses BRDF auto-encoders, and Paint-it (Youwang et al., 2024) re-parameterizes PBR textures for efficient synthesis. More recently, 3DTopia (Chen et al., 2025) uses 2D diffusion priors to refine 3D model textures through latent and pixel space optimization for high-quality material generation. Material Anything (Huang et al., 2025) focuses specifically on the material generation problem, utilizing a pre-trained image diffusion model with a triple-head architecture enhanced by a rendering loss which allows for fully automated material generation for any 3D object. However, a core challenge persists in these methods: the inherent ambiguity of text prompts in conveying the complex and nuanced appearance of materials, which limits the precision of the final output.

**Image-based Generation.** To overcome text limitations, recent work uses images as more precise visual guidance. Zero-1-to-3 (Liu et al., 2023) fine-tunes pre-trained 2D diffusion models for novel view synthesis from a single image, but suffers from multi-view inconsistency due to independent view generation. One-2-3-45 (Liu et al., 2024) addresses this by reconstructing 3D objects from predicted multi-view images using generalizable NeRF, improving geometric consistency. However,

these methods retain implicit material representations that lack tileability and standard editability, making them difficult to export and transfer across different models.

Another line of work leverages large multimodal models for material retrieval. (Fang et al., 2024) employs GPT-4V as a material analyzer to automate the generation of physically-based SVBRDFs for 3D assets. By relying on retrieval, however, this approach is limited to the materials available in its database, and it cannot generate novel materials or offer fine-grained control over material parameters.

PHYSICS-BASED INVERSE RENDERING

**Physics-Based Differentiable Rendering.** To accurately recover material parameters from images, differentiable rendering is an essential tool. Physics-based differentiable rendering (PBDR) combines modern, physically-principled rendering techniques with automatic differentiation to provide a powerful framework for inverse rendering problems. However, standard automatic differentiation of Monte Carlo estimators produces biased gradients when applied to complex light transport phenomena involving discontinuities from visibility events. To address this fundamental issue, (Li et al., 2018) first introduced edge sampling for Monte Carlo ray tracing, enabling unbiased differentiation of the rendering equation (Kajiya, 1986). They further generalized this into the path-space differentiable rendering (PSDR) framework (Zhang, 2022), which operates directly on the path integral formulation and traces discontinuities along light paths for improved performance. (Loubet et al., 2019) introduced an approximate re-parameterization technique to bypass the explicit sampling of discontinuities, which integrated into differentiable renderer Mitsuba 3 (Jakob et al., 2022).

**Inverse Procedural Materials.** A burgeoning area within PBDR is the recovery of procedural materials, which are highly valued for their resolution independence and editability. Early methods relied on hand-crafted templates and rules to extract structural and local information from source exemplars. For more stochastic textures, procedural noise models were optimized to match the appearance by fitting their power spectrum in either the image or frequency domain. With the advent of node-based material authoring tools, a new branch of methods emerged to match the appearance of photorealistic materials. MATch (Shi et al., 2020) introduced DiffMat, a library that converts large-scale procedural material graphs into a differentiable format, enabling end-to-end, gradient-based optimization of material parameters to match a target appearance. As an improvement, (Hu et al., 2022) developed differentiable neural proxies to approximate non-differentiable generator nodes, allowing for the joint optimization of both continuous and discrete parameters. Similarly, recent systems like PSDR-Room (Yan et al., 2023) and Mapa (Zhang et al., 2024) have combined differentiable procedural materials with PBDR for scene-level material reconstruction and text-driven material painting, respectively.

## 2 METHOD

We propose a three-stage framework (Figure 1) that progressively refines materials from coarse approximations to precise PBR representations: Our framework consists of: (1) structure-guided appearance transfer using generative models, (2) hybrid material initialization combining procedural retrieval with texture generation, and (3) differentiable optimization for precise appearance matching.

### 2.1 STAGE 1: STRUCTURE-GUIDED APPEARANCE TRANSFER

A fundamental obstacle in single-image material acquisition is the domain gap between the reference image's viewpoint and the target 3D geometry. To circumvent this, we formulate the problem as a task of *image-to-image translation editing*. To this end, we employ a state-of-the-art, prompt-guided image editing model (e.g., GPT-Image-1), selected for its superior ability to disentangle and transfer complex material attributes like material, texture, color, and reflectance from a source image.

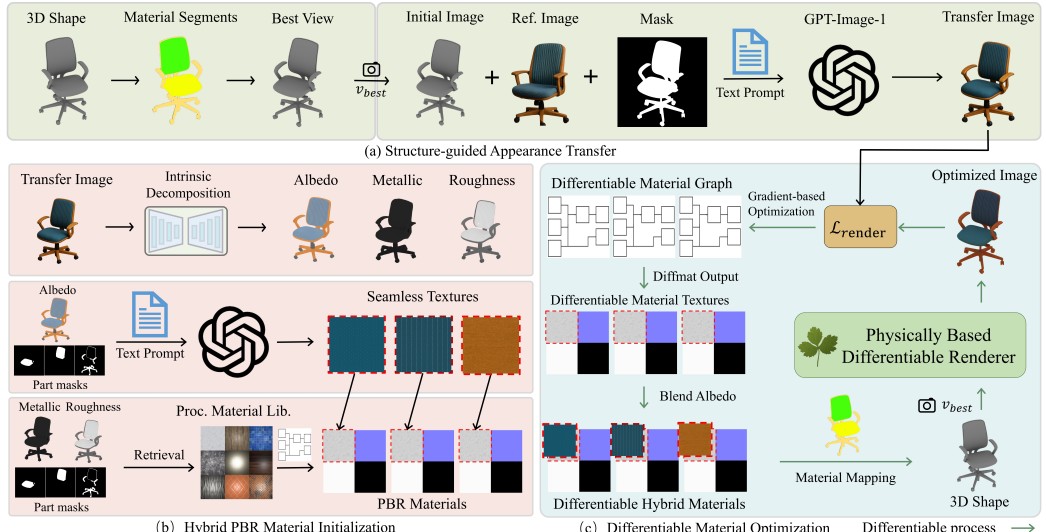

Figure 1: Our three-stage pipeline for generating PBR materials from a single image. (a) We synthesize a geometrically-aligned target image that transfers the reference style. (b) A hybrid material is initialized by blending a procedural base with a generated seamless texture. (c) The material is then optimized within a differentiable renderer to match the target.

### 2.1.1 MULTI-MODAL INPUT CONSTRUCTION.

To address the challenge of viewpoint selection in single-image material transfer, we develop a systematic approach to find an optimal viewpoint that maximizes the visibility of all material regions. Our approach operates on a 3D model $S$ partitioned into $K$ material segments, $\{S_k\}_{k=1}^{K}$. For the models used in our experiments, we leverage their predefined material groups. The number of segments $K$ is therefore inherited from the source model's structure. For assets lacking such groups, an automatic mesh segmentation method like that of SAMesh (Tang et al., 2024) can be applied as a preprocessing step. With these segments defined, our two-stage process for finding the optimal viewpoint is as follows.

First, we sample $n$ candidate viewpoints $v_i{}_{i=1}^{n}$ on a sphere surrounding the model. Our objective is to find the views that see the maximum number of distinct material segments. For each viewpoint $v_i$, we render a segmentation mask $M_i$ and simply count the number of unique segments visible:

$$K_i = |\{k \in \{1, \ldots, K\} : S_k \cap M_i \neq \emptyset\}| \tag{1}$$

After computing this for all candidates, we create a subset of viewpoints, $V_{\text{candidate}}$, containing only those that achieve the maximum count ($K_{\max}$).

Second, from this candidate set, we select the single best viewpoint. A good viewpoint should not only see many segments but also provide **balanced visibility**, avoiding cases where some segments are occluded or shown as only a few pixels. To enforce this, we apply a "maximin" criterion: we find the viewpoint that maximizes the area of its *least* visible segment. First, for each candidate view $v_j$, we find the area of its smallest visible part, $\min_{S_k \in \mathcal{S}_j} \text{Area}(S_k \cap M_j)$. Then, we select the viewpoint that maximizes this value:

$$v_{\text{best}} = \arg \max_{v_j \in V_{\text{candidate}}} \left( \min_{S_k \in \mathcal{S}_j} \text{Area}(S_k \cap M_j) \right) \tag{2}$$

This criterion ensures that all visible material regions are substantially represented. From this optimal viewpoint $v_{\text{best}}$, we render a clean, untextured image of the model's geometry, denoted as $I_{\text{geom}}$, which serves as the structural template for subsequent appearance transfer.

### 2.1.2 STRUCTURE-GUIDED IMAGE EDITING.

We formulate the appearance transfer as a controlled image-to-image translation task. The editing model receives three inputs: the reference style image $I_{\text{ref}}$, the target structure image $I_{\text{geom}}$ ren-

dered from the optimal viewpoint, and a carefully designed prompt that enforces strict structural constraints while enabling style transfer.

The prompt instructs the model to perform a stylized rendering that adheres to the following requirements:

- *"Strictly preserve the structure and geometric shape of the original model from Image 1."*
- *"The visual style must fully match the material, color, and texture characteristics of Image 2."*
- *"The output should exhibit the artistic style of Image 2 while retaining structural details from Image 1."*

The output of this process is an edited image, $I_{\text{edit}}$. To ensure geometric alignment, we then generate the final transfer image, $I_{\text{tr}}$, by projecting the style from $I_{\text{edit}}$ onto our target geometry. This is achieved using a mask of the model rendered from the optimal viewpoint, $v_{\text{best}}$. This resulting image, $I_{\text{tr}}$, serves as the crucial, pixel-aligned supervision signal for our subsequent material optimization pipeline.

## 2.2 STAGE 2: HYBRID PBR MATERIAL INITIALIZATION

With the supervision image $I_{tr}$ established, we devise a hybrid strategy to initialize the material for each surface segment, addressing the fundamental limitation of procedural material libraries: While procedural materials offer physical plausibility and editability, material libraries are inherently limited in their coverage of real-world appearances. Direct retrieval from such libraries often fails to capture the diverse textures and patterns present in arbitrary reference images. Pure optimization from a poorly-matched initial material typically converges to suboptimal local minima. Therefore, we enhance the retrieved base material with generative texture synthesis.

### 2.2.1 INTRINSIC IMAGE DECOMPOSITION

Before material retrieval, we decompose the transferred image $I_{tr}$ to extract its surface reflectance properties using RGB-X (Zeng et al., 2024), a pre-trained diffusion model trained on over 200,000 paired samples. The decomposition process yields:

$$D(I_{tr}) = (I_{tr}^{\text{alb}}, I_{tr}^{\text{nor}}, I_{tr}^{\text{rgh}}, I_{tr}^{\text{met}}) \tag{3}$$

where $I_{tr}^{\text{alb}}$, $I_{tr}^{\text{nor}}$, $I_{tr}^{\text{rgh}}$, and $I_{tr}^{\text{met}}$ represent the extracted albedo, normal, roughness, and metallic maps respectively. These decomposed BRDF properties provide physically-meaningful features for subsequent material matching.

### 2.2.2 PHYSICS-BASED MATERIAL RETRIEVAL

With the decomposed BRDF properties from $I_{tr}$, we perform physics-informed material retrieval for each material segment. For each segment $S_k$ and each candidate procedural material $P_m$ from our library $\mathcal{P}$, we map its physical parameter textures (roughness $\theta_{\text{rgh}}^m$ and metallic $\theta_{\text{met}}^m$) onto the geometry via UV coordinates. We then render these mapped parameters from the optimal viewpoint, where pixel intensities directly correspond to the parameter values.

The matching is performed by comparing these projected parameter maps with the corresponding regions in the decomposed maps:

$$P_k^* = \arg \min_{P_m \in \mathcal{P}} \left[ \lambda_{\text{rgh}} \| I_{tr,k}^{\text{rgh}} - P_k(\theta_{\text{rgh}}^m) \|_2^2 + \lambda_{\text{met}} \| I_{tr,k}^{\text{met}} - P_k(\theta_{\text{met}}^m) \|_2^2 \right] \tag{4}$$

where $P_k(\cdot)$ denotes an operator that projects the UV-mapped parameter textures of segment $S_k$ into screen space, and $I_{tr,k}^{\text{rgh}}$, $I_{tr,k}^{\text{met}}$ represent the decomposed roughness and metallic values in the region corresponding to segment $S_k$. For small material segments where texture details are less critical, we assign uniform materials without retrieval to maintain computational efficiency.

### 2.2.3 GENERATIVE TEXTURE SYNTHESIS AND BLENDING

To capture unique texture patterns absent from our library, we leverage the same image editing model to generate tileable textures for each material segment. For each segment $S_k$, we extract its corresponding masked region from $I_{tr}$ and provide it to the model with the following prompt:

- *"Minimize texture distortion to ensure proper tiling."*
- *"Preserve the color and pattern characteristics."*

This yields a segment-specific albedo texture $T_{A,k}^{\text{gen}}$ that captures the visual patterns for material segment $S_k$. We then blend this texture with the albedo map $T_{A,k}^{\text{proc}}$ from the retrieved procedural material:

$$T_{A,k}^{\text{final}} = (1 - w_k) \cdot T_{A,k}^{\text{proc}} + w_k \cdot T_{A,k}^{\text{gen}}, \tag{5}$$

where $w_k \in [0,1]$ is a learnable blending weight for segment $S_k$. This hybrid approach combines the structural coherence and physical plausibility of procedural materials with the visual richness of generated textures, providing a robust initialization for each material segment in the subsequent optimization stage.

### 2.3  STAGE 3: DIFFERENTIABLE MATERIAL OPTIMIZATION

The final stage optimizes all learnable parameters to precisely match the supervision image $I_{tr}$. Given the high-dimensional nature of the optimization problem, we adopt a coarse-to-fine strategy that progressively refines different parameter groups.

**Optimization Parameters.**  Our method optimizes three groups of parameters: the environment map $E$ representing scene illumination; the UV transformation parameters $\theta_{\text{uv}} = \{s, r, t\}$ for scale, rotation, and translation; and the material parameters $\theta_{\text{mat}}$, which include the procedural parameters $\theta_{\text{proc}}$ and the blending weight $w$.

**Progressive Optimization Strategy.**  We decompose the optimization into three sequential steps, each focusing on different aspects of the appearance:

*Step 1: Lighting Estimation.* We first optimize the environment map $E$ while keeping all material parameters fixed:

$$\hat{E} = \arg \min_{E} \mathcal{L}_{\text{render}}(R(S, \theta_{\text{init}}, E), I_{tr}) \tag{6}$$

This establishes a plausible lighting condition that explains the overall illumination in the reference image.

*Step 2: UV Alignment.* We initialize the texture blending weight $w = 1$ to ensure clear pattern matching. For UV transformation, we first evaluate a discrete set of rotation and scale combinations, selecting the best initialization based on VGG Gram matric distance (Gatys et al., 2016). With lighting fixed to $\hat{E}$, we then optimize the UV transformation parameters:

$$\hat{\theta}_{\text{uv}} = \arg \min_{\theta_{\text{uv}}} \mathcal{L}_{\text{VGG}}(R(S, \theta_{\text{mat}}, \theta_{\text{uv}}, \hat{E}), I_{tr}) \tag{7}$$

This ensures that texture scales and orientations match the reference appearance.

*Step 3: Joint Material Refinement.* Finally, we jointly optimize material parameters and fine-tune UV coordinates:

$$(\hat{\theta}_{\text{mat}}, \hat{\theta}_{\text{uv}}) = \arg \min_{\theta_{\text{mat}}, \theta_{\text{uv}}} \mathcal{L}_{\text{render}}(R(S, \theta_{\text{mat}}, \theta_{\text{uv}}, \hat{E}), I_{tr}) \tag{8}$$

**Loss Function.**  The rendering loss combines global appearance matching with segment-specific perceptual terms. Let $I_{\text{render}} = R(S, \theta_{\text{mat}}, \theta_{\text{uv}}, E)$ denote the final rendered image produced by our differentiable renderer $R$. The overall loss is defined as:

$$\mathcal{L}_{\text{total}} = \mathcal{L}_{\text{render}} + \sum_{k=1}^{K} \lambda_k \mathcal{L}_{\text{VGG}}^{k}(I_{\text{render},k}, I_{tr,k}) \tag{9}$$

where $\mathcal{L}_{\text{render}} = \lambda_{\text{global}} \|D(I_{\text{render}}) - D(I_{tr})\|_1$ is an L1 loss that captures global color consistency, and $D(\cdot)$ is the downsampling operator.

The term $\mathcal{L}_{\text{VGG}}^{k}$ is a segment-specific perceptual loss computed on the image patches corresponding to each material segment $S_k$. Let $I_{\text{render},k}$ and $I_{tr,k}$ be the image patches extracted from $I_{\text{render}}$ and

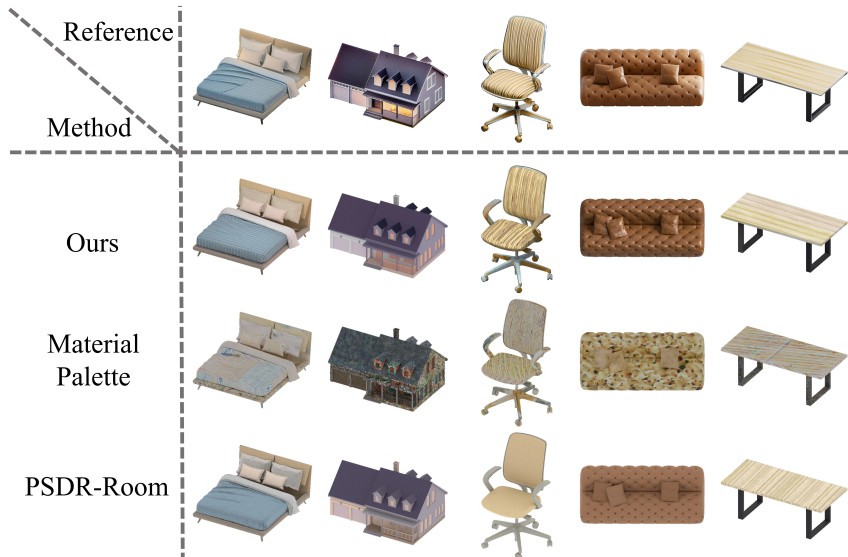

Figure 2: Qualitative comparison of material generation methods. For fair comparison, all baseline methods use our Stage 1 transferred image as input, as they lack structure-guided appearance transfer capabilities.

$I_{tr}$ respectively, using the mask for segment $S_k$. The loss for each segment is:

$$\mathcal{L}^k_{\text{VGG}}(I_{\text{render},k}, I_{tr,k}) = \sum_{l \in \mathcal{F}} \frac{\lambda_l}{C_l^2} \|\mathcal{G}_l(I_{\text{render},k}) - \mathcal{G}_l(I_{tr,k})\|_1 \tag{10}$$

where $\mathcal{G}_l(\cdot)$ computes the Gram matrix of features from layer $l$ of a pre-trained VGG network. Here, $\phi_l(\cdot) \in \mathbb{R}^{C_l \times H_l W_l}$ extracts and reshapes features, $\mathcal{F}$ denotes the set of selected feature layers, and $C_l, H_l, W_l$ are the channel, height, and width dimensions. This per-segment perceptual loss captures fine-grained texture and style similarities specific to each material region.

## 3 EXPERIMENTS

**Differentiable Renderer and Optimization.** Our physics-based optimization pipeline is built upon the **Mitsuba 3** differentiable renderer (Jakob et al., 2022). For propagating gradients through the rendering process, we utilize the **reparameterization** technique of Loubet et al. (Loubet et al., 2019), which handles discontinuities from visibility events. We employ a **path tracing integrator** for all rendering operations. The rendering resolution is set to 512×512 pixels. To generate high-quality, anti-aliased images, we use a a indipendent sampler generates 64 samples-per-pixel (spp).

### 3.1 QUALITATIVE EVALUATION

We evaluate our method on diverse 3D models across six categories and compare against two baselines on imaged based appearance modeling: PSDR-Room (Yan et al., 2023), Material and Palette (Lopes et al., 2024).

**Comparison with Baselines.** Figure 2 compares our method against two strong baselines. PSDR-Room, limited by its procedural-only approach, fails to capture fine-grained appearance variations. Material Palette's SVBRDF extraction suffers from resolution dependencies and lacks physical constraints, resulting in spatially incoherent texture synthesis when input resolution falls below 4K. In contrast, our hybrid representation effectively balances visual fidelity with physical plausibility, achieving superior reconstruction quality across diverse material categories.

**Material Generation Fidelity.** Figure 3 presents our final rendering results across diverse object categories, demonstrating high-fidelity reconstruction of various material properties from leather to

Figure 3: Material generation results across diverse object categories.

Figure 4: Multi-view evaluation shows consistent appearance.

Table 1: Quantitative evaluation of rendering quality. We compare rendered images of 3D shapes with generated materials against transfer images. Higher PSNR/SSIM and lower LPIPS indicate better visual fidelity. Best results are in **bold**.

| Shape Category | Ours | | | PSDR-Room | | | Material Palette | | |
|---|---|---|---|---|---|---|---|---|---|
| | PSNR↑ | SSIM↑ | LPIPS↓ | PSNR↑ | SSIM↑ | LPIPS↓ | PSNR↑ | SSIM↑ | LPIPS↓ |
| All | **16.549** | **0.575** | **0.160** | 16.199 | 0.484 | 0.182 | 13.071 | 0.456 | 0.203 |
| Bags | 15.499 | 0.455 | 0.193 | **16.614** | **0.516** | 0.193 | 13.562 | 0.469 | 0.223 |
| Beds | **17.440** | **0.658** | **0.172** | 16.854 | 0.460 | 0.216 | 14.845 | 0.554 | 0.204 |
| Buildings | **14.239** | **0.395** | 0.220 | 14.162 | 0.367 | 0.225 | 12.823 | 0.209 | **0.247** |
| Chairs | **15.232** | **0.461** | **0.128** | 15.187 | 0.413 | 0.153 | 13.459 | 0.376 | 0.150 |
| Sofas | **15.723** | **0.640** | **0.175** | 15.030 | 0.535 | 0.195 | 12.893 | 0.542 | 0.224 |
| Tables | **18.666** | **0.623** | **0.124** | 18.051 | 0.499 | 0.149 | 12.304 | 0.441 | 0.182 |

wood. In these examples, our hybrid approach combines physically-correct procedural properties with generated texture details to achieve a realistic appearance that remains faithful to the reference. We provide additional results in the appendix, including the application of diverse material styles to a single geometry (see Figure 8).

**Multi-View Consistency.** Figure 4 validates that our single-view optimization produces materials that generalize well to novel viewpoints. This robustness stems from our physics-based representation: procedural materials inherently encode view-dependent effects correctly, while our differentiable optimization enforces physical constraints throughout the process. This addresses a key limitation of many appearance transfer methods that produce view-dependent artifacts.

## 3.2 QUANTITATIVE EVALUATION

We quantitatively evaluate our method against baselines using standard metrics for material reconstruction quality.

**Rendering Quality Assessment.** Table 1 measures the visual quality of rendered results after applying generated materials to 3D shapes. Our method produces renderings that best match reference images across all metrics. The superior PSNR (16.549) and SSIM (0.575) demonstrate accurate reproduction of visual appearance, while the lowest LPIPS score (0.160) indicates our renderings are perceptually closest to target images. Strong performance on furniture categories reflects our method's ability to capture complex appearance variations that define visual realism.

**User Study.** To evaluate perceptual fidelity, we conducted a user study. We created 54 unique test cases by selecting 3 representative models and 3 stylistically diverse reference images from each of our 6 object categories. We recruited 30 participants to perform a blind test comparing our method against two baselines (PSDR-Room and Material Palette). In each trial, participants were shown a reference image alongside the three rendered results, presented in a randomized order to prevent bias. They were asked to rate each result on a 5-point Likert scale (1=Poor, 5=Excellent) based on how well it replicated the reference material's appearance (including texture, color, and reflectance).

Table 2: User study results. We report the average user preference scores on a 5-point scale (1=Poor, 5=Excellent). Each row represents a material generation method.

| Method | Shape Category | | | | | | Overall Avg. |
|---|---|---|---|---|---|---|---|
| | Bags | Beds | Buildings | Chairs | Sofas | Tables | |
| **Ours** | **3.56** | **4.07** | **3.96** | **3.80** | **4.20** | **3.83** | **3.90** |
| PSDR-Room | 3.37 | 3.32 | 3.53 | 3.40 | 3.17 | 3.44 | 3.37 |
| Material Palette | 2.77 | 1.95 | 1.78 | 2.38 | 2.53 | 2.20 | 2.27 |

The results in Table 2, show that our method achieved the highest score, indicating a strong user preference.

### 3.3 ABLATION STUDY

To validate our core contribution, the hybrid material representation, we conduct an ablation study comparing two configurations:

- **W/O textures:** Optimizes only the parameters of the retrieved procedural material.
- **Ours:** Our full pipeline, which jointly optimizes the procedural material and a generated texture using a learnable blending weight.

Figure 5 illustrates the comparison on a multi-material chair. The **W/O textures** method adjusts parameters to match the reference's color palette, such as the hue of the wood and fabric. However, being constrained by the base material's definition, it fails to synthesize patterns not present in the original procedure. In contrast, **Ours** approach seamlessly integrates

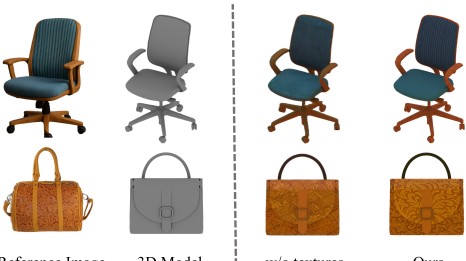

Reference Image    3D Model    w/o textures    Ours

Figure 5: Ablation study comparison. W/O textures optimizes to match the reflectance but cannot match the textures. Our captures both reflectance and textures, validating our hybrid approach.

these details by blending in the generated texture. This successfully reproduces the fine-grained patterns, achieving a result faithful in both physical properties and detailed appearance.

## 4 CONCLUSION

We presented Style2Shape, a framework for generating high-fidelity materials for 3D shapes from a single reference image. Our multi-stage design directly addresses the ill-posed nature of this task. By first performing **structure-guided appearance transfer**, we bridge the domain gap to create an aligned supervision target. Subsequently, our **hybrid PBR material initialization** provides a robust starting point by combining the *reflectance correctness* of procedural materials with generated textures. This enables our physics-based optimization to converge to solutions that are editable, physically plausible, and visually faithful, as validated by our experiments.

**Limitations and Future Work.** While our framework shows promising results, it has several limitations. First, our experiments primarily focus on transferring materials where the reference image and target shape share a similar semantic category. The framework's performance on cross-category transfers with large semantic gaps (e.g., applying a sofa texture to a dragon) has not been evaluated and remains a challenging area for future work. Second, as observed in some supplementary examples, performance on real-world photos with complex, non-uniform lighting can be suboptimal, as the initial BRDF decomposition may be less accurate. Third, our hybrid blending mechanism currently operates only on the albedo channel to maximize appearance matching. Extending this hybrid approach to other PBR maps like roughness and metallic, while ensuring physical plausibility, is a non-trivial but important future direction. Finally, our Stage 1 depends on a powerful, proprietary image editing model; exploring open-source alternatives to improve accessibility is also a key next step.

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

# A APPENDIX

## A.1 ADDITIONAL RESULTS GALLERY

We present an extended collection of material generation results that demonstrate the robustness and versatility of our method. These examples complement the main paper by showcasing our framework's performance on a broader range of 3D models and reference materials, including challenging generalization tests.

**Multi-Model, Multi-Category Variation.** Figure 6 and Figure 7 demonstrate our method's capability to handle geometric and material diversity across various models and object categories (e.g., tables, beds, sofas, bags). Our framework successfully adapts to each geometry while faithfully reproducing both global appearance and fine-grained surface properties.

**Single-Model, Multi-Style Versatility.** A key capability of our framework is its versatility in applying different material styles to a single geometry, enabling efficient design exploration. As shown in Figure 8, we can take one chair model and apply a range of distinct appearances—from fabric to leather to wood—each derived from a different reference image. This highlights the ability of our pipeline to consistently generate high-quality materials regardless of the target style.

**Cross-Category Generalization.** To further investigate the framework's generalization capabilities, we conducted experiments on challenging **cross-category material transfers**. Figure 9 showcases two such results. The examples illustrate the successful application of a wooden texture and a marble texture from table reference onto a chair model (left), and a leather and a fabric pattern from handbag onto a table (right). This demonstrates the versatility of our approach, which effectively transfers complex material attributes across significant semantic gaps while maintaining visual coherence and key pattern details.

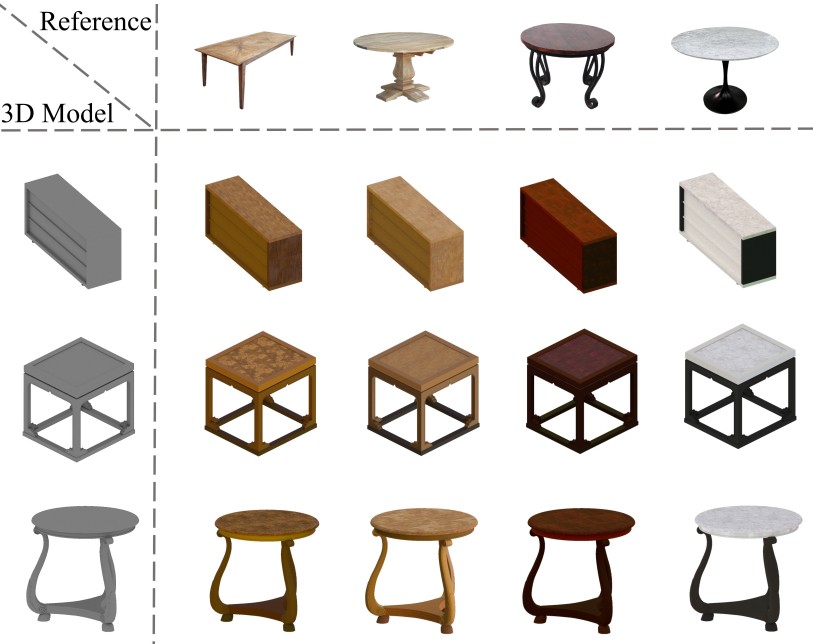

Figure 6: Material generation results within the tables category.

## A.2 A DEEPER LOOK INTO THE PIPELINE

To provide further insight into our framework, we visualize and discuss key intermediate outputs and design choices that are critical to the method's success.

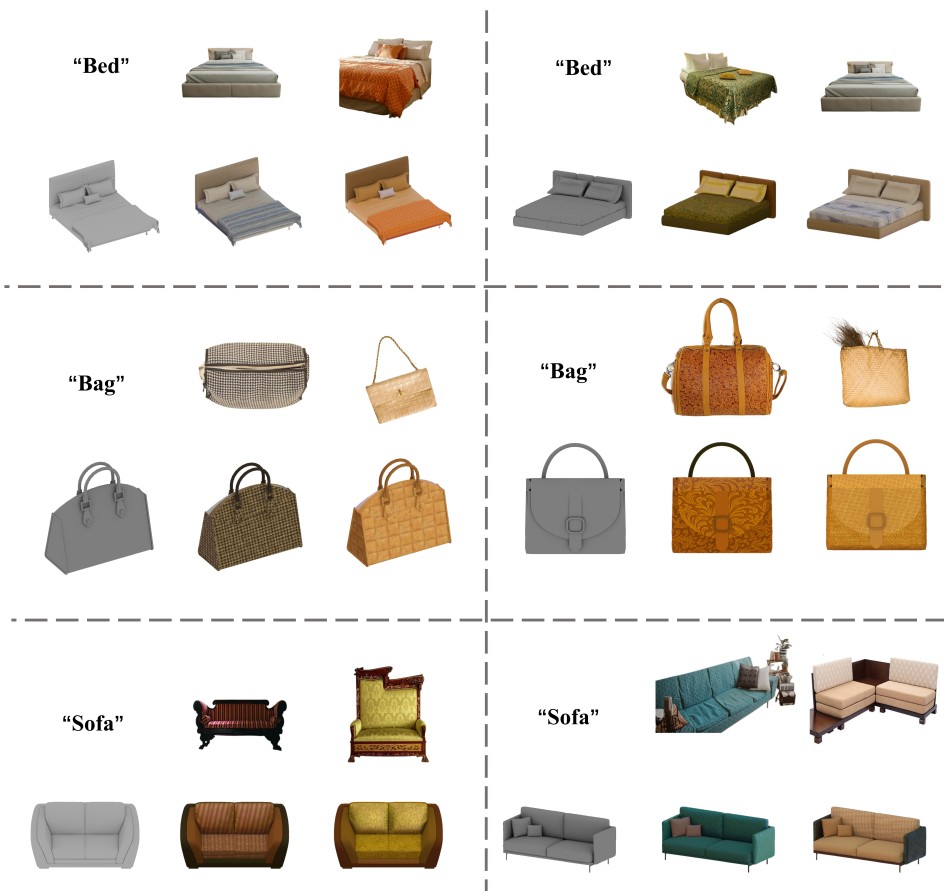

Figure 7: Material generation results on multiple categories (beds, sofas, bags) paired with varied reference materials.

**The Importance of Structure-Guided Transfer.** A core challenge is the domain gap between a 2D reference image and a 3D target shape. A naive approach might simply project the reference texture, leading to severe distortions. Our Stage 1 circumvents this by generating a structure-guided transfer image ($I_{tr}$). As shown in Figure 10, our structured prompts successfully guide the image editing model to transfer material properties while preserving the geometric integrity of the target model. This process yields a clean, pixel-aligned supervision signal that has the reference style but conforms perfectly to the target geometry and viewpoint, which is crucial for the subsequent physics-based optimization to converge correctly.

**Hybrid Material Components.** Figure 11 displays the generated albedo textures ($T_{A,k}^{\text{gen}}$) from Stage 2. These textures, created with prompts requesting seamless patterns, are designed to capture intricate details from the reference style. They work in synergy with the retrieved procedural base material; the procedural component provides a physically coherent foundation for reflectance, while the generated texture provides the specific visual patterns. Their successful integration via our optimization framework validates the effectiveness of our hybrid material representation.

A.3 ANALYSIS OF GENERATED PBR MATERIALS

**BRDF Map Coherence.** Figure 12 presents a comprehensive analysis of our generated PBR materials through their constituent BRDF maps. The albedo maps reveal the successful integration of our hybrid approach—base procedural colors are seamlessly blended with generated texture details. In contrast, the roughness, metallic, and normal maps are derived directly from the optimized procedural material. This design choice ensures that these physical parameters retain their inherent

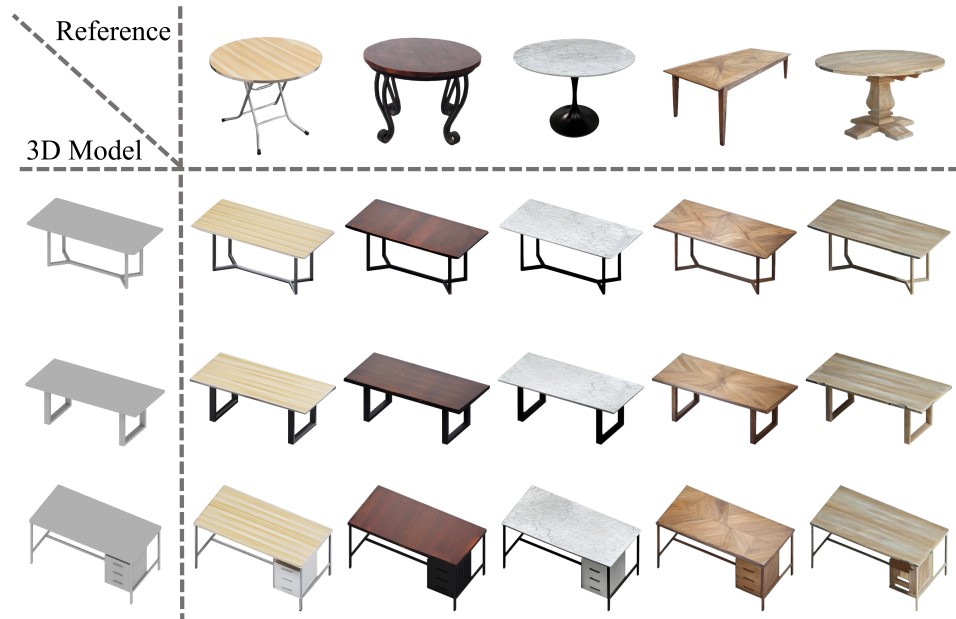

Figure 8: **Versatility on a single object with diverse reference images.** Our method enables applying multiple distinct material appearances to the same 3D geometry while maintaining consistent material quality across different styles. This is ideal for rapid design exploration.

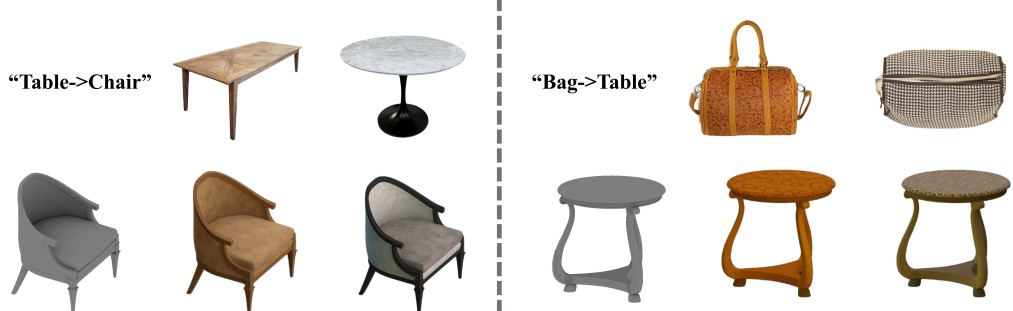

Figure 9: **Demonstration of Cross-Category Material Transfer.** Our framework successfully transfers material styles between objects of disparate semantic categories. The examples show transfers from a table to a chair (left panels) and from a handbag to a table (right panels).

coherence from the procedural graph, accurately controlling surface reflectance. These complete parameter sets confirm that our optimization process produces materials with plausible light interaction properties, faithful to the reference's appearance.

A.4 IMPLEMENTATION DETAILS

**Dataset and Material Library.** Our 3D model dataset is sourced from 3DCoMPaT (Yuchen Li, 2022) and 3D Warehouse (Trimble Inc., 2024), covering six categories: bags, beds, buildings, chairs, tables, and sofas, with at least 5 models per category. Reference images consist of real-world photographs under CC0 license, similarly spanning all six categories with at least 5 images per category. For material matching and optimization, we collect 263 base procedural materials from Substance Designer (Adobe Inc., 2024) in .sbs format. We sample 5 parameter variants for each base material by varying key parameters such as roughness, scale, and color, resulting in a total library of 1,315 material instances exported at 512×512 resolution.

**Differentiable Renderer and Optimization.** Our physics-based optimization pipeline is built upon the **Mitsuba 3** differentiable renderer (Jakob et al., 2022). For propagating gradients through

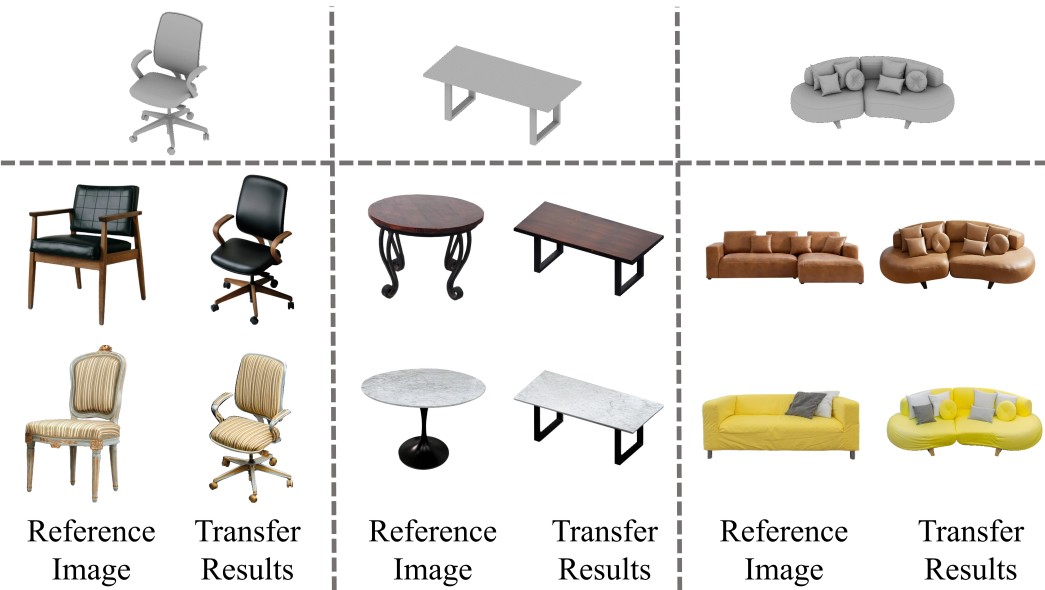

| Reference Image | Transfer Results | Reference Image | Transfer Results | Reference Image | Transfer Results |
|---|---|---|---|---|---|

Figure 10: **Structure-guided appearance transfer results (Stage 1).** This figure showcases the output of the first stage of our pipeline. We leverage a powerful image editing model with structured prompts to transfer material properties from a source image while preserving the geometric integrity of diverse target models. The resulting images serve as the supervision targets for our optimization.

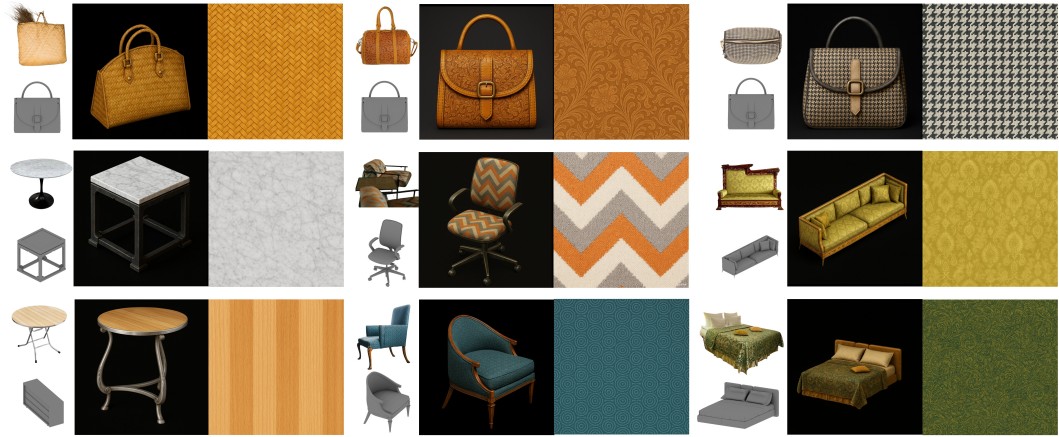

Figure 11: Visualization of pipeline intermediate results. Left: Structure-guided transfer images ($I_{tr}$) from Stage 1 providing pixel-aligned supervision. Right: Generated albedo textures ($T_{A,k}^{\text{gen}}$) from Stage 2 capturing detailed patterns for the hybrid material representation.

the rendering process, we utilize the **reparameterization** technique of Loubet et al. (Loubet et al., 2019), which handles discontinuities from visibility events. We employ a **path tracing integrator** for all rendering operations. The rendering resolution is set to 512×512 pixels. To generate high-quality, anti-aliased images, we use a indipendent sampler generates 64 samples-per-pixel (spp). We initialize the environment map $E$ as a 256×256 HDR image with uniform unit radiance and apply 8× downsampling during the rendering loss computation. The perceptual loss employs a pre-trained VGG-16 network. For UV transformation initialization, we evaluate discrete candidates with rotation angles $r \in \{0, \pi/4, \pi/2, -\pi/2, -\pi/4\}$ and scale factors $s \in \{0.5, 1.0, 2.0, 4.0, 8.0\}$. We employ a three-stage optimization schedule using the Adam optimizer (Kingma & Ba, 2014): (i) 50 iterations for lighting estimation with lr = $10^{-2}$, where we apply a weak **total variation regularizer** to the environment map $E$ to encourage spatial smoothness; (ii) 50 iterations for UV alignment with lr = $10^{-3}$; and (iii) 100 iterations for joint refinement with cosine annealing from lr = $10^{-2}$ to

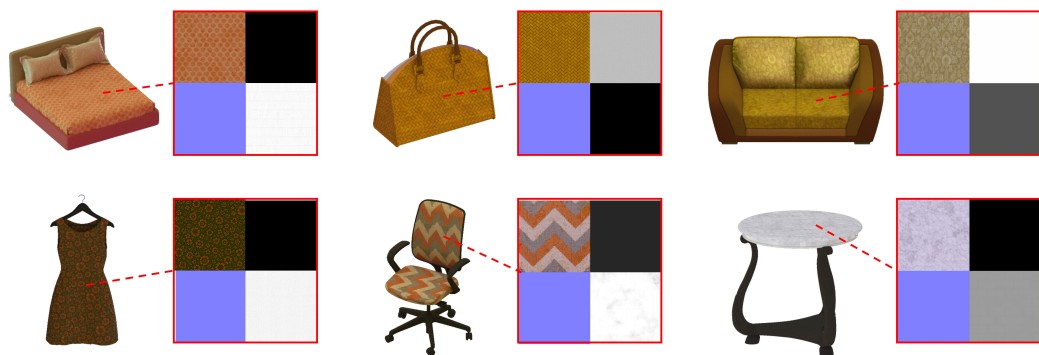

Figure 12: Visualization of generated PBR materials. For each rendered result (left), we show the complete set of BRDF parameter maps(right): Albedo, Roughness, Metallic, and Normal. The physical coherence and visual quality of these maps demonstrate our method produces PBR materials well-structured for integration into existing workflows.

$10^{-4}$ for material parameters. The complete pipeline requires approximately 10 minutes on a single NVIDIA RTX 3090 GPU.

