# OpenReview forum: "Style2Shape: Image Style Guided 3D Shape Material Generation"
_ICLR.cc/2026/Conference — Submitted to ICLR 2026_

### Official Review · Reviewer_oWHJ · 2025-10-28

**Soundness:** 2
**Presentation:** 2
**Contribution:** 2
**Rating:** 2
**Confidence:** 4

**Summary:**

Style2Shape is a framework for generating PBR materials for 3D models conditioned on a reference image. It combines procedural materials with generative techniques to balance physical plausibility and visual fidelity. The pipeline has three stages: (1) structure-guided appearance transfer, (2) hybrid PBR material initialization, and (3) differentiable material optimization. The method is compared against two baselines, PSDR-Room and Material Palette, and reports superior performance.

**Strengths:**

1. Clear motivation. Procedural materials provide physical correctness but have limited expressivity, while generative textures are expressive but not guaranteed to be physically valid. The proposed hybrid approach is well motivated.
2. Empirical results indicate better visual fidelity than the baselines.

**Weaknesses:**

1. Several relevant texture generation methods are missing, for example TexGaussian and Meta3DGen.
2. The paper compares only to two procedural methods, omitting many generative approaches with the same problem setting, for example RomanTex, Hunyuan3D-2.0, TEXGen, SyncMVD, and Paint3D. Without these, the evaluation is not fully convincing.
3. The ablation focuses on incorporating generated textures, leaving other design choices under explained. For example, how does the VGG term contribute in Exp(9)? How much does the Progressive Optimization Strategy help compared to direct optimization?
4. The paper claims preprocessing via SAMesh extends to objects beyond the six categories, but no substantive examples are provided, aside from the dress in Fig. 12 which lacks distinct material parts. In addition, semantic segmentation does not guarantee material homogeneity within segments, and it may not merge spatially separated regions of the same material. This limits generalization.
5. The method assumes an optimal viewpoint that covers all material regions. This can hold for simple objects such as tables and chairs, but often fails for complex geometry such as human characters and garments, which constrains applicability.
6. Section indices are inconsistent. There is no index before RELATED WORK, and MATERIAL GENERATION FOR 3D OBJECT is incorrectly labeled as 1.1.

TexGaussian: Generating High-quality PBR Material via Octree-based 3D Gaussian Splatting
Meta 3d texturegen: Fast and consistent texture generation for 3d objects
RomanTex: Decoupling 3D-aware Rotary Positional Embedded Multi-Attention Network for Texture Synthesis
Hunyuan3d 2.0: Scaling diffusion models for high resolution textured 3d assets generation
TEXGen: a Generative Diffusion Model for Mesh Textures
Text-guided texturing by synchronized multi-view diffusion.
Paint3D: Paint Anything 3D with Lighting-Less Texture Diffusion Models

**Questions:**

How many objects were used for quantitative evaluation, and is this the same set used in the user study?

In Table 1, why does performance degrade for the bag category?

“For small material segments where texture details are less critical, we assign uniform materials without retrieval to maintain computational efficiency.” What uniform materials are assigned in practice, and how are their parameters chosen?

---

### Official Review · Reviewer_rigK · 2025-11-01

**Soundness:** 1
**Presentation:** 2
**Contribution:** 2
**Rating:** 4
**Confidence:** 3

**Summary:**

TLDR: A complex, three-stage framework generating PBR materials from single image.

This paper proposes Style2Shape, which generates PBR materials for 3D models from a single reference image. Extending PSDR-Room (retrieves procedural materials and optimizes parameters), this work adds texture generation via image editing models to capture fine-grained appearances. However, the approach assumes albedo and roughness/metallic are independent (problematic), relies on multiple external dependencies, and uses multiiple ad-hoc components that create a fragile cascade.

**Strengths:**

- Tries to combine image generated textures with procedural patterns.
- Transfer results look interesting.

**Weaknesses:**

- Overly complex and fragile cascade system that have multiple ad-hoc components - failure in any stage causes irreversible harm. Three sequential stages with heavy external dependencies (parts segmentation, image editing model, RGB-X decomposition, procedural library, differentiable renderer).
- Problematic independence assumption: Stage 2 separately retrieves roughness/metallic and generates albedo, but in reality these are correlated. Inconsistency between these two components will make the material not physically plausible. And to really see if the reflectance make sense, we need to see a view-varying video.
- Assumes simplistic textures - method may not generalize to other type of textures.
- "Editability" claimed but not demonstrated - no actual editing examples
- Have to find the best pose of the 3D shape so the image can capture most of the parts

**Questions:**

1. How this work will inspire and stimulate future work in this direction?

---

### Official Review · Reviewer_RvFp · 2025-11-02

**Soundness:** 3
**Presentation:** 4
**Contribution:** 3
**Rating:** 6
**Confidence:** 4

**Summary:**

This paper introduces Style2Shape, a novel and comprehensive framework for generating editable, physically-based rendering (PBR) materials for a 3D model from a single reference image. The core contribution is a hybrid material representation that synergistically combines procedural materials and AI-generated textures. The authors argue that this approach leverages the strengths of both: procedural materials ensure physical correctness of reflectance properties (e.g., roughness, metallicity), while generative models capture arbitrary and complex visual appearances from the reference.

**Strengths:**

The central contribution—a learnable blend of procedural materials and generated textures—is a powerful and elegant idea. It effectively combines the strengths of both paradigms, achieving results that are simultaneously physically plausible and visually faithful to an arbitrary reference.

The three-stage pipeline is very well-designed. Stage 1 intelligently solves the critical domain gap problem. Stage 2 provides a strong initialization that is crucial for the success of the high-dimensional optimization. Stage 3 uses a progressive optimization strategy to robustly converge to a high-quality solution. This structured approach demonstrates a deep understanding of the problem's complexities.

A major strength is that the final output consists of standard, editable PBR texture maps. This makes the generated assets directly usable in modern 3D engine or rendering software.

**Weaknesses:**

My comments primarily echo and expand upon those points, framing them as areas for improvement.

The proposed "maximin" criterion for optimal viewpoint selection is logical, but it does not address scenarios where some material segments are inherently occluded from any single viewpoint (e.g., the inside of a cup, the underside of a complex chair). The paper does not specify how materials are generated for segments that are not visible from the chosen optimal view. This omission represents a potential failure point for complex geometries.

The success of Stage 1 heavily relies on a state-of-the-art, prompt-guided image editing model (referred to as "GPT-Image-1"). The paper would be strengthened by demonstrating that the framework is robust to the choice of the image model, for instance, by showing results with open-source alternatives like ControlNet or InstructPix2Pix.

The framework's material retrieval and subsequent optimization are contingent on the quality of the decomposed BRDF maps from Stage

2. The paper cites RGB-X, but in practice, single-image intrinsic decomposition models often struggle with complex, non-uniform lighting, strong cast shadows, or highly specular surfaces. The real-world performance of such models can be suboptimal, which could lead to inaccurate initialization of roughness and metallic properties, potentially hindering the optimization process or leading to physically incorrect results. The paper would benefit from a discussion on how errors from this decomposition stage propagate and how the framework might mitigate them.

**Questions:**

Could the authors clarify the specific model used for the structure-guided image editing? Have they experimented with publicly available models (e.g., based on Stable Diffusion with ControlNet)? How sensitive are the final results to the quality of the image generated in this first stage?

The viewpoint selection method aims to maximize visibility, but for complex models, it's inevitable that some material segments will remain unseen from any single optimal viewpoint. How does the framework handle material generation for these occluded parts? Are they assigned a default material, do they inherit properties from adjacent visible segments, or is there another mechanism in place?

Given that models like RGB-X can produce artifacts or inaccurate maps under challenging lighting, how does the system perform when the initial Irgh and Imet maps are noisy or incorrect? Does the final differentiable optimization stage have the capacity to correct for a poor initialization stemming from decomposition errors, or does it typically converge to a suboptimal result?

---

### Official Review · Reviewer_Hn2v · 2025-11-03

**Soundness:** 3
**Presentation:** 3
**Contribution:** 2
**Rating:** 2
**Confidence:** 4

**Summary:**

This paper introduces Style2Shape, a framework designed to generate materials from a single reference image. The pipeline comprises three stages: (1) aligning 3D geometry with the reference image, (2) initializing material parameters through a retrieval mechanism, and (3) refining procedural materials via differentiable rendering. The system is engineering-complete, and the presented results demonstrate the effectiveness.

However,  the core modules and overall pipeline strongly overlap with MaPa (SIGGRAPH 2024): both use 2D diffusion generation as a bridge to produce aligned images, both rely on a procedural material library for retrieval, and both refine procedural parameters with differentiable rendering. The primary distinction lies in the modality of input that Style2Shape accepts the image as input, while MaPa is conditioned on text.  The authors do not clearly validate its unique advantages or contributions over MaPa.

While the system is well-engineered, the overlap in methodology limits its originality. I think the contribution may fall short of the innovation threshold expected at ICLR.

**Strengths:**

- A complete three-stage pipeline that jointly addresses multiple subproblems in alignment, initialization, and refinement

**Weaknesses:**

- Novelty

  The paper has high overlap with MaPa and lacks direct comparison or justification of unique contributions.

- Clarity

  The paper’s exposition could be improved. For instance, in line 43, it points out the drawbacks of procedural material generation, yet in line 46, it begins to praise its advantages, which creates a logical inconsistency and may confuse readers.

- Experiments

  1. The paper proposes a three-stage pipeline, yet lacks corresponding ablation studies to verify the importance of each component. For example, there is no experiment showing the effect of misalignment when the input image is not properly aligned with the geometry.
  2. In line 36, the authors mention the issue of specular ambiguity as a motivation for introducing procedural modeling, yet the presented results do not include materials with strong specular highlights, such as metals, it is not sure the performance of the proposed method for such cases.
  3. The paper lacks comparison to several relevant methods, such as MaterialMVP (ICCV 2025), which also supports image as the condition and generates the object material. A comparative analysis would be beneficial to highlight the strengths of the proposed formulation, making the paper more solid

- Figure

  Some of the text in the figure is low-resolution and difficult to read. The authors are encouraged to improve the rendering quality or resolution

- Technical limitation

The paper attempts to use text prompts to generate seamless patterns. However, it cannot guarantee true seamlessness with only text prompts constraints.

**Questions:**

1. In the supplementary materials, is Style2Shape/images/bag-005/seamless_0 intended to be the output SVBRDF folder? If so, it is evident that the normal map in this directory is incorrect. Moreover, the result is not truly seamless, as I mentioned.

2. The normal maps depicted in Fig. 12 appear excessively smooth. It is unclear whether this is due to a downsampled resolution in the visualization or if the predicted normals themselves inherently lack high-frequency detail.

3. Most of the results are demonstrated on furniture models with relatively simple geometry. Is that possible to apply the method to the complex model and other types (like a metallic dragon)?

---

### Meta-Review · Area_Chair_xr7S · 2026-01-03

**Summary:**

Reviewers pointed out that the proposed pipeline is overly complex. The reviewers have concerns with the novelty issues due to overlap with MaPa and lack a clear justification of contributions. Reviewers also suggested additional experimental validation and ablation studies.
Finally, there are also issues with clarity, e.g., low-resolution figures.

**Reviewer Concerns:**

The authors did not provide a rebuttal. Hence, all the concerns are outstanding.

**Reviewer Scores:**

The authors did not provide a rebuttal. Therefore, the reviewers would not change their score even if they had been able to participate in the discussion.

---

### Decision · Program_Chairs · 2026-01-26

Reject